# The Mechanism of Action and Clinical Efficacy of Low-Dose Long-Term Macrolide Therapy in Chronic Rhinosinusitis

**DOI:** 10.3390/ijms24119489

**Published:** 2023-05-30

**Authors:** Gwanghui Ryu, Eunkyu Lee, Song I Park, Minhae Park, Sang Duk Hong, Yong Gi Jung, Hyo Yeol Kim

**Affiliations:** 1Department of Otorhinolaryngology-Head and Neck Surgery, Samsung Medical Center, Sungkyunkwan University School of Medicine, Seoul 06351, Republic of Korea; 2Department of Otorhinolaryngology-Head and Neck Surgery, Ilsan Paik Hospital, Inje University College of Medicine, Goyang 10380, Republic of Korea

**Keywords:** chronic rhinosinusitis, macrolide, cytokines, neutrophils, nasal polyps

## Abstract

Various chronic inflammatory airway diseases can be treated with low-dose, long-term (LDLT) macrolide therapy. LDLT macrolides can be one of the therapeutic options for chronic rhinosinusitis (CRS) due to their immunomodulatory and anti-inflammatory actions. Currently, various immunomodulatory mechanisms of the LDLT macrolide treatment have been reported, as well as their antimicrobial properties. Several mechanisms have already been identified in CRS, including reduced cytokines such as interleukin (IL)-8, IL-6, IL-1β, tumor necrosis factor-α, transforming growth factor-β, inhibition of neutrophil recruitment, decreased mucus secretion, and increased mucociliary transport. Although some evidence of effectiveness for CRS has been published, the efficacy of this therapy has been inconsistent across clinical studies. LDLT macrolides are generally believed to act on the non-type 2 inflammatory endotype of CRS. However, the effectiveness of LDLT macrolide treatment in CRS is still controversial. Here, we reviewed the immunological mechanisms related to CRS in LDLT macrolide therapy and the treatment effects according to the clinical situation of CRS.

## 1. Introduction

Low-dose long-term (LDLT) macrolide therapy is a type of treatment in which the dosage is lower than that used to treat an acute bacterial infection and the duration is longer than that normally used. The regimen was first reported for the treatment of patients with diffuse panbronchiolitis with LDLT erythromycin in Japan in 1984 [1]. Since then, it has been widely used for chronic airway diseases such as chronic obstructive pulmonary disease (COPD), asthma, diffuse panbronchiolitis, bronchiectasis, cystic fibrosis, and idiopathic pulmonary fibrosis [2]. LDLT macrolide therapy has been found to enhance lung function and reduce the frequency and severity of exacerbations in people with these conditions [3]. It is thought that the immunomodulatory and anti-inflammatory potency of macrolides, through various mechanisms, can effectively control these diseases, as can their antimicrobial properties [4]. In addition, macrolide antibiotics have been reported to have therapeutic potential through immunomodulation in a variety of different diseases, such as rheumatoid arthritis, coronary artery disease, non-small cell lung cancer, periodontitis, and blepharitis [5,6].

Chronic rhinosinusitis (CRS) is one of the chronic inflammatory diseases of the upper respiratory tract. CRS has a similar pathophysiology to the above-mentioned lower airway inflammatory diseases, particularly asthma [7,8]. In an early study in 1970, macrolide therapy was able to reduce corticosteroid doses in patients with asthma [9]. A systematic review of the effects of long-term macrolide treatment on asthma found that the treatment reduced exacerbations and symptoms but did not significantly increase lung function [10]. In a multicenter randomized controlled trial (RCT) conducted in patients with severe asthma in Belgium, LDLT azithromycin reduced severe exacerbations and lower respiratory tract infections only in patients with non-eosinophilic severe asthma [11]. Therefore, the authors proposed a role for LDLT macrolides in severe asthmatic patients with corticosteroid insensitivity.

CRS is classified into CRS with nasal polyps (CRSwNP) and CRS without nasal polyps (CRSsNP) according to the phenotype and is divided into type 2 and non-type 2 according to the endotype [12]. As in asthma, corticosteroid therapy in CRS is effective for type 2 and CRSwNP patients, but some cases do not respond to it [13]. Eosinophilic CRSwNP was associated with higher type 2 cytokine expression, such as interleukin (IL)-5, IL-13, and eotaxin. On the other hand, non-type 2 CRS had more neutrophilic inflammation and IL-8 [14]. According to the results of studies about CRS endotypes and biomarkers, the medical treatment of CRS requires customization by patients, including corticosteroids, antibiotics, and biologicals [15]. Appropriate medical therapy for CRS includes a combination of treatments such as intranasal corticosteroid spray, short-term oral corticosteroids, and nasal saline irrigation. If medical treatment is ineffective, functional endoscopic sinus surgery (FESS) can be recommended [12]. Still, the role of antibiotics, including LDLT macrolides, in the treatment of CRS remains controversial.

In general, LDLT macrolide for the treatment of CRS is known to act on non-type 2 inflammation with low levels of eosinophils and immunoglobulin (Ig) E [16]. The treatment with LDLT macrolides in CRS was introduced by a Japanese group in 1991 [17]. Moriyama et al. reported that post-operative LDLT erythromycin showed better improvement of clinical symptoms and endoscopic findings compared to the non-treated group [18]. Since then, LDLT macrolides, including 14-membered lactone erythromycin, roxithromycin, clarithromycin, and 15-membered azithromycin [19], have been studied in various clinical trials and have been frequently prescribed to patients with CRS [20,21]. The European Position Paper on Rhinosinusitis and Nasal Polyps (EPOS) 2012 guidelines recommended LDLT macrolide treatment for CRSsNP patients with normal total IgE levels [22]. So far, this treatment has been known to be effective only in the non-type 2 endotype of CRS [23]. However, its use in CRS is not recommended in the EPOS 2020 guidelines [12]. The reason is that there are no large-scale RCTs on CRS and no specific studies on the clinical phenotype and endotype of CRS [24]. However, the prescription of macrolide-based treatment for CRS is emerging, and the frequency of prescription is high in actual clinical practice [25]. Macrolide antibiotics are the second most prescribed drug after penicillin/beta-lactams in the United States and are also preferred by doctors at university hospitals in South Korea [20,21].

In this review, we investigated the immunological mechanisms related to CRS of LDLT macrolide treatment, focusing on the therapeutic effect according to the clinical situation of CRS. In particular, the evidence of LDLT macrolide therapy was summarized based on the phenotype (CRSwNP and CRSsNP) and endotype of CRS (type 2 vs. non-type 2). Additionally, the duration of treatment, use in children, and side effects of LDLT macrolides were reviewed. Unless otherwise noted, all clinical studies were conducted on adult patients with CRS.

## 2. Mechanism of Action of LDLT Macrolide Therapy in CRS

Immunomodulation effects of LDLT macrolides in chronic airway disease are driven by multiple pathways, including cytokine and chemokine production, cellular recruitment, mucus secretion, barrier function, biofilm formation, and tissue fibrosis [2,4]. The effects of LDLT macrolide are mainly known to decrease type 1 cytokines and reduce neutrophil recruitment. However, macrolides have also been reported to reduce type 2 cytokines such as IL-4 and IL-5 in chronic airway disease [6]. Treatment with LDLT macrolides has also been found to affect mucociliary clearance and epithelial barrier function [26]. These effects may play a role in the pathophysiology of CRS (Figure 1).

### 2.1. Reducing Proinflammatory Cytokines

The major immune regulatory effect of macrolides is to reduce the production of proinflammatory cytokines in various inflammatory cells. Macrolides decrease the production of IL-6 and tumor necrosis factor (TNF)-α [27,28]. Azithromycin inhibits the inflammasome and reduces IL-1β secretion in monocytes and macrophages [29,30]. These inhibitory effects are regulated by the alteration of cellular signaling pathways, such as mitogen-activated protein kinase (MAPK), extracellular signal-regulated kinase 1/2 (ERK1/2), and nuclear factor (NF)-κB [4].

In CRSsNP patients, the nasal mucosa was cultured with clarithromycin, and the secretion of IL-5, IL-8, and granulocyte-macrophage colony-stimulating factor (GM-CSF) was decreased [31]. In addition, transforming growth factor (TGF)-β and NF-κB were decreased when nasal mucosal tissues were treated with clarithromycin [32]. However, the results were inconsistent in human samples treated with 250 mg of clarithromycin for three months.

After treatment with clarithromycin for eight weeks in CRSwNP patients, levels of IL-6, IL-8, and IL-1β in the nasal secretions were reduced [33]. Another study published by the same author showed decreased eosinophilic inflammatory markers, including regulated on activation, normal T cell expressed and secreted (RANTES), and eosinophilic cationic protein (ECP), after eight weeks of clarithromycin treatment [34]. In an in vitro study, erythromycin suppressed the production of eotaxin and RANTES in a lung fibroblast cell line (human fetal lung fibroblasts 1) [35]. Postoperative clarithromycin treatment significantly reduced ECP levels in nasal secretion at 12 and 24 weeks, but not in the control group [36]. However, conflicting results were found, with no difference in ECP level of nasal secretions between LDLT erythromycin and the placebo group [37].

### 2.2. Inhibition of Neutrophil Recruitment

IL-8, also known as C-X-C Motif Chemokine Ligand 8 (CXCL8), has been identified as a function of neutrophil recruitment. Erythromycin can inhibit the production of IL-8 by neutrophils and eosinophils [38,39]. Previously, Suzuki et al. reported that the administration of roxithromycin to patients with CRS reduced neutrophil counts and IL-8 levels in nasal secretion [40]. This effect was also confirmed by an RCT with 64 CRSsNP patients [41].

Reduced production of IL-8 and IL-1β can block the extravascular transmigration of neutrophils through inhibition of transcription factors such as NF-κB and activator protein-1 (AP-1) [42]. In healthy subjects and COPD patients, short-term administration of azithromycin reduced IL-8 and soluble vascular cell adhesion molecule (VCAM)-1 and modulated neutrophil function [43,44]. Furthermore, azithromycin suppressed the proliferation and cytokine production of CD4^+^ T cells, especially IL-17 secretion via the mammalian target of the rapamycin (mTOR) pathway [45].

### 2.3. Mucus Secretion and Mucociliary Clearance

Macrolides can reduce the expression of MUC5AC in airway epithelial cells [26]. Clarithromycin and erythromycin effectively inhibited the expression of MUC5AC in human nasal epithelial cells from CRSwNP patients [46]. Azithromycin also significantly reduced MUC5AC expression in human nasal epithelial cells [47]. In rats stimulated with intratracheal lipopolysaccharide (LPS), roxithromycin treatment significantly reduced Muc5ac expression and NF-κB nuclear translocation in the bronchial epithelium [48]. Azithromycin and clarithromycin showed the same effect in ovalbumin (OVA)-sensitized and LPS-instilled rats [46,47]. In human bronchial epithelial cells, clarithromycin inhibited the expression of MUC5AC and IL-13-induced goblet cell hyperplasia [49,50]. Similar to other inhibitory mechanisms, clarithromycin had an impact on NF-κB inactivation. In a CRS mouse model, the level of IL-10 was increased, and Muc5ac expression was inhibited by erythromycin treatment [51].

In patients with acute purulent rhinitis, clarithromycin treatment for two weeks reduced secretion volume and increased mucociliary transportability [52]. Clarithromycin had the same effect of significantly reducing mucus viscosity and nasal clearance time in CRS patients treated for four weeks [53,54]. In a three-month RCT in patients with CRS, saccharine transit time was significantly improved in the roxithromycin group compared to the placebo group [41]. Improvement in mucociliary clearance, as measured by saccharine transit time, persisted after 12 months of follow-up [55].

### 2.4. Epithelial Barrier Function

Asgrimsson et al. reported that azithromycin, but not erythromycin, induced expression of tight junction proteins, including claudin-1, claudin-4, occludin, and junctional adhesion molecule-A, and increased epithelial integrity in human bronchial epithelial cells [56]. During *Pseudomonas aeruginosa* infection in vitro, pretreatment with azithromycin prevented epithelial barrier dysfunction and enhanced recovery [57]. The potential protective effects of macrolides on human respiratory epithelium were investigated in vitro [58,59]. Macrolides, such as roxithromycin, clarithromycin, and azithromycin, reduced the production of reactive oxygen species generated by activated neutrophils [58]. These agents were able to attenuate the injurious effects of bioactive phospholipids and neutrophil-induced epithelial damage [59]. Lastly, roxithromycin treatment increased ciliary movement and mucociliary transport velocity in the rabbit trachea [60].

### 2.5. Inhibition of Biofilm Formation

Biofilms are a surrounding structure of microorganisms that can provide resistance to host immune responses and antimicrobial agents and are also important for CRS pathophysiology [61]. Bacterial biofilms induced by *Staphylococcus aureus* or *Pseudomonas aeruginosa* contribute to the severity and refractoriness of CRS [62]. Korkmaz et al. reported the biofilm eradication effect of eight weeks of clarithromycin treatment in a RCT in CRSwNP patients [63]. Compared to the mometasone furoate nasal spray group (1 of 11), there were more patients (6 of 12) with biofilm disappearance in the clarithromycin treatment group. Previous in vitro studies found that macrolides can inhibit the production of bacterial proteins and reduce biofilm formation by *Pseudomonas aeruginosa* [64,65]. Recently, antibiotic (ciprofloxacin and azithromycin)-eluting sinus stents have been experimentally demonstrated to inhibit *Pseudomonas aeruginosa*-induced biofilms [66]. The authors demonstrated that prolonged release of ciprofloxacin and azithromycin for 28 days reduced biofilm formation and eliminated existing biofilms.

### 2.6. Effects on Tissue Fibrosis

Several in vitro studies have shown that macrolides inhibit fibroblasts in nasal polyps. When nasal polyp-derived fibroblasts were treated with roxithromycin and then stimulated with lipopolysaccharide, fibroblast proliferation was inhibited [67]. This suppression phenomenon was actually observed in the fibroblasts of CRSwNP patients treated with roxithromycin for one month [68]. In addition, roxithromycin inhibited the production of nitric oxide [69], IL-6 and RANTES [69], matrix metalloproteinase (MMP)-2, and MMP-9 [70] in TNF-α-stimulated nasal polyp fibroblasts. Another in vitro study with nasal polyp fibroblasts demonstrated that erythromycin and roxithromycin treatment reduced TGF-β-induced α-smooth muscle actin (a myofibroblast marker), collagen production, nicotinamide adenine dinucleotide phosphate oxidase 4, and reactive oxygen species production [71]. Collectively, these findings indicate an inhibitory effect of macrolide treatment on fibroblast-induced nasal polyp formation and may explain the mechanism of polyp size reduction in patients with CRSwNP.

## 3. Comparison of Clinical Efficacy in CRS

Although there are few direct comparative studies of each clinical situation, RCT studies of macrolides are summarized. Most of the studies compared placebo with macrolides (Table 1), and some compared treatment with conventional CRS treatment, intranasal corticosteroid spray (Table 2). Herein, the results of clinical trials were comprehensively reviewed according to the presence or absence of nasal polyps, type of inflammation, total IgE level, and the presence or absence of allergy.

### 3.1. CRSwNP vs. CRSsNP

After long-term clarithromycin treatment (8 to 12 weeks) in 20 CRSwNP patients, 40% of patients had a reduction in nasal polyp size and a significant decrease in IL-8 levels in lavage fluid, while 60% remained unchanged [81]. Preoperative treatment with 500 mg of clarithromycin for eight weeks reduced polyp recurrence at 6 and 12 months postoperatively [77]. Computed tomography (CT) findings and SNOT-20 improved in CRSwNP patients treated with mometasone furoate monotherapy and LDLT clarithromycin combination therapy for eight weeks, but there was no statistically significant difference between the two groups [63]. In 52 CRSwNP patients treated with LDLT clarithromycin for 12 weeks, there were significant reductions in the Sinonasal Outcome Test (SNOT)-20 and Lund-Kennedy endoscopy score [82]. In addition, 54% (28 of 52) of those who improved on SNOT-20 had lower total IgE levels than others.

In CRSsNP patients, treatment with 150 mg roxithromycin for three months showed significant improvement in sinonasal symptoms (SNOT-20), nasal endoscopy findings, and mucociliary transit time [41]. When comparing the effects of mometasone furoate and LDLT clarithromycin at three months, there was no significant difference in visual analog scales of symptoms or endoscopic findings between the two groups [80]. Eight weeks of erythromycin treatment also showed clinical improvement in CRSsNP patients [83].

However, in a mixed cohort of CRSwNPs (52.0%) and CRSsNPs, except for severe polyposis, LDLT azithromycin did not differ between treatment and placebo groups [78]. Treatment with LDLT azithromycin for three months after FESS in CRS patients improved SNOT-22 compared to conventional treatment [76], whereas erythromycin treatment after FESS was ineffective in CRSwNP (55.2%) and CRSsNP [37]. There was no additional effect of clarithromycin for three months with budesonide aqua nasal spray in patients with CRS (56.8% with nasal polyps) [84]. Haruna et al. retrospectively analyzed patients who received LDLT macrolides treatment for 8–20 weeks; the clinical effect was good in CRSsNP patients, whereas the effect increased after polypectomy in CRSwNP patients [85]. In a biomarker study for the prediction of the macrolide treatment group in patients with CRS postoperatively, nasal tissue IgG4 level and overall symptom score were identified as predictive factors for refractoriness [86]. However, there was no difference in refractory rate between the LDLT clarithromycin treatment group (18 of 74) and the fluticasone propionate spray group (17 of 75).

### 3.2. Type 2 vs. Non-Type 2

Treatment with macrolides (clarithromycin or roxithromycin) for 2–3 months improved clinical symptoms in CRS patients, and the degree of clinical improvement was inversely correlated with eosinophil counts in the peripheral blood, the nasal smear, and the sinus mucosa [87]. However, the number of neutrophils, mast cells, and mononuclear cells did not correlate with symptomatic improvement, and the number of interferon-γ and IL-4-positive cells also did not correlate.

Zeng et al. compared the efficacy of fluticasone propionate nasal spray versus LDLT clarithromycin for postoperative treatment in CRS of different phenotypes. The study found that both medications were effective in reducing symptoms, but there were no significant differences between eosinophilic (>10% eosinophils/total infiltrating cells) and non-eosinophilic CRSwNP groups [79]. Asians, who are generally known to have more non-type 2 CRS, showed better treatment effects of LDLT macrolides than non-Asians in a meta-analysis [88].

A recent study showed that long-term treatment with clarithromycin was effective in CRSwNP patients without tissue eosinophilia (>10 eosinophils/high power field) [74]. When comparing oral steroids alone with oral steroids plus clarithromycin for 12 weeks in CRSwNP patients who underwent FESS, symptom scores and endoscopy scores improved significantly in the add-on treatment group [74]. In a case-control study of LDLT clarithromycin after surgery, responders (19 of 28, 67.9%) had lower blood eosinophil counts (0.16 ± 0.11 versus 0.39 ± 0.36 10^9^/L) and tissue eosinophilia (>10 eosinophils/high power field, 17.6% versus 62.5%) compared to non-responders [89]. According to these studies, patients with type 2 inflammation of CRS have a lower response to LDLT macrolide therapy.

Aspirin or nonsteroidal anti-inflammatory drug (NSAID)-exacerbated respiratory disease (AERD/NERD) is characterized by asthma, CRSwNP, and aspirin or nonsteroidal anti-inflammatory drug intolerance [90]. In AERD patients with eosinophilic nasal polyps (>40% eosinophils), LDLT azithromycin treatment significantly reduced symptoms (visual analog scale and SNOT-22) and the need for surgery (74% versus 14%) compared to placebo [75]. In addition, another study showed that azithromycin significantly improved disease clearance in AERD patients compared to placebo [73]. In patients with refractory CRS who failed surgery and medical treatment, azithromycin treatment not only alleviated symptoms but also significantly reduced the amount of *Staphylococcus aureus* [72,73]. These recent studies have demonstrated that LDLT macrolide treatment is also effective in CRS patients with eosinophilic inflammation.

### 3.3. Normal vs. High Total IgE

Previous studies reported that only CRS patients with normal serum IgE levels (<200 μg/L [41] or ≤250 U/mL [87]) benefited from LDLT macrolide treatment. However, the relationship between total serum IgE levels and LDLT macrolide treatment effects is still controversial. According to the studies showing that LDLT macrolide treatment was effective, the total serum IgE in the patient group was 188.63 ± 57.25 IU/mL [77] and 165.0 ± 195.2 μL/L [37]. Maniakas et al. reported that total serum IgE was higher in the azithromycin success group compared to the azithromycin failure group [91]. In addition, atopy status did not affect the clinical effect of clarithromycin in CRSsNP patients [80].

Recently, of the 100 CRS patients who were administered LDLT roxithromycin, 29 were determined to be responders [92]. Among clinical parameters, including nasal secretion and serum IgE, IL-5, blood eosinophil/neutrophil, allergy, asthma, and nasal polyps, total IgE in nasal secretions was the only predictor of responder in multivariate models (odds ratio 4.76, 95% confidential interval 1.29–17.58). The authors suggest that local total IgE is a reliable biomarker instead of serum total IgE.

### 3.4. Allergic vs. Non-Allergic Patients

Yamada et al. evaluated the effectiveness of LDLT clarithromycin in patients with non-allergic CRSwNP and found a significant reduction in nasal polyp size and IL-8 secretion in 40% (8 of 20) of patients [81]. CRS patients with or without allergies have different responses to treatment with LDLT macrolides. In patients with confirmed CRSwNP allergy status by skin prick test, ECP levels in nasal secretions decreased in allergic patients, and IL-6 levels decreased only in allergic patients after eight weeks of clarithromycin treatment [34]. However, allergic status had no impact on the clinical efficacy of LDLT macrolides [85,93].

## 4. Other Considerations

There are several factors to consider while prescribing LDLT macrolides, including the type of medication, dose, duration, and timing of treatment. Patient characteristics such as age, underlying disease, and comorbidities should be considered in the treatment. In addition, we should be aware of the adverse effects of long-term treatment. Although there has been no well-designed study that directly compared the treatment effects between macrolides in CRS, there was a recent study comparing the effects of two drugs. Comparing the effects of clarithromycin and azithromycin treatment for four weeks, azithromycin was more effective for complete resolution of symptoms and CT scores [94]. In a systematic review and meta-analysis of clarithromycin in CRS compared with the intranasal corticosteroid spray, there was no significant difference in effectiveness [95]. However, combined treatment with clarithromycin and intranasal corticosteroid spray markedly improved clinical symptoms, endoscopic findings, and Lund-Mackay CT scores.

### 4.1. Duration of Treatment

LDLT macrolide treatment is known to be more effective the longer the treatment period. After treatment with 8 to 12 weeks of clarithromycin in CRS patients, symptoms and endoscopic findings improved in 71.1% of participants, and the clinical effect was correlated with the duration of treatment [96]. In a meta-analysis, the effects were more favorable in patients taking LDLT macrolides for 24 weeks in comparison to 8 and 12 weeks [97]. Treatment with clarithromycin for 24 weeks after FESS resulted in better CT scores compared to those for 12 weeks [36]. Treatment with clarithromycin showed clinical effects after four weeks and reached its maximum effect at 12 weeks in patients with CRSsNP [80]. Nakamura et al. compared the clinical efficacy of LDLT clarithromycin in patients with CRS postoperatively. In the 6-month treatment group, the rate of asymptomatic improvement was higher at 12 months after surgery than in the 3-month treatment group [98]. Taken together, the longer the treatment period, the better the clinical outcome of LDLT macrolides.

### 4.2. Pediatric CRS Patients

Some evidence for LDLT macrolide treatment has also been reported in pediatric patients with CRS [99]. A retrospective review of six patients (mean age: 7 ± 3.4 years) who were treated with either roxithromycin or clarithromycin found that macrolide add-on therapy improved nasal symptoms and reduced thick mucus secretions [100]. After administration of clarithromycin at a half dose (5–8 mg/kg) for eight weeks to 54 children with CRS, 63.0% were cured and 31.5% were improved [101].

Therapeutic effects of LDLT macrolides, such as improving lung function and reducing exacerbations, have been demonstrated in chronic inflammatory diseases of the lower respiratory tract, such as severe asthma and cystic fibrosis in children [99]. Unfortunately, no randomized, placebo-controlled clinical trials of LDLT treatment in children have been conducted.

### 4.3. Adverse Effects of LDLT Macrolides

During LDLT macrolide treatment for 8 to 12 weeks, there is no strong evidence of the development of drug-resistant bacteria strains [16]. However, LDLT azithromycin use over 12 to 24 months in pediatric patients with bronchiectasis resulted in an increased presence of macrolide-resistant organisms [102]. Macrolides also carry a risk of prolongation of the QT interval and consequent torsades de pointes arrhythmia [103,104]. On the other hand, the incidence of torsades de pointes with erythromycin was very rare (four cases out of 34,000 patients treated) [105]. In a national cohort that included 66,331 CRS patients, the risk of mortality and cardiovascular events was not significantly increased in patients who had been prescribed macrolides, particularly clarithromycin, compared to penicillin [106]. Clarithromycin treatment was known to increase the risk of stroke and myocardial infarction, but a nation-wide cohort study showed no association with overall mortality or long-term cardiovascular death [107,108]. Nonetheless, caution is required if the patient is at risk of a cardiac event prior to the initiation of LDLT macrolide treatment [16].

## 5. Conclusions

Because CRS is a highly heterogeneous disease entity, the clinical efficacy of LDLT macrolide therapy is variable. Several RCTs have demonstrated that LDLT macrolides can improve symptoms and quality of life in patients with CRS, particularly those with CRSsNP, normal total IgE levels, and corticosteroid resistance [109]. The immunomodulatory and anti-inflammatory properties of macrolides contribute not only to the reduction of neutrophilic inflammation but also to the decrease of eosinophilic inflammation, mucus clearance, and mucosal stabilization. In addition, it can increase the effectiveness of treatment by removing bacterial biofilm and preventing or reducing polyp formation by inhibiting tissue fibrosis. These various mechanisms may have an impact on CRS treatment for numerous clinical conditions. The efficacy of LDLT macrolide therapy may be influenced by the endotype and phenotype of CRS.

Further research is needed to fully understand the mechanisms underlying the therapeutic effect of macrolides in CRS and to identify the most appropriate patients for this treatment approach, including non-antibiotic macrolides [2,110]. Currently, non-antibiotic macrolides such as EM900, an erythromycin derivative, are being developed and researched and may be spotlighted as an important treatment modality for CRS in the future [111]. Nonetheless, current evidence suggests that low-dose, long-term macrolide therapy is a promising option for the management of CRS. LDLT macrolide treatment may be the main treatment for certain subtypes of CRS and may be used as an additional treatment with corticosteroids for other types of CRS.

## Figures and Tables

**Figure 1 ijms-24-09489-f001:**
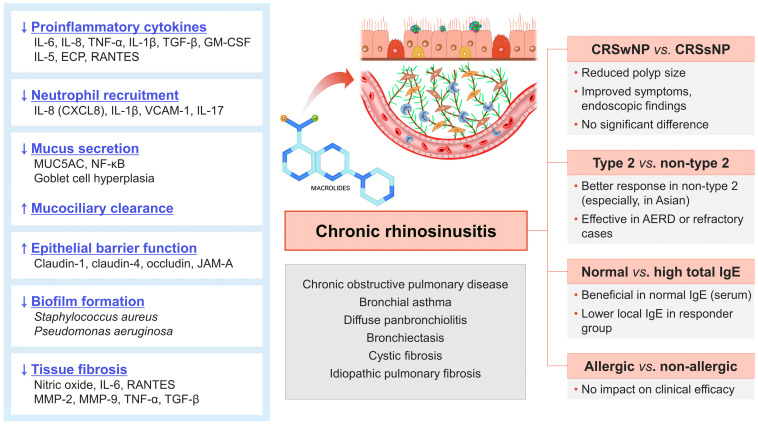
Mechanism of action and clinical implications of low-dose, long-term macrolide therapy in chronic rhinosinusitis.

**Table 1 ijms-24-09489-t001:** Randomized placebo-controlled trials using low-dose long-term macrolides in chronic rhinosinusitis.

Reference	Patients	Number	Nasal Polyp	Macrolide	Duration
Renteria, 2021 [72]	Refractory CRS post-FESS	48	91.7%	Azithromycin 250 mg 3 times per week	4 months
Maniakas, 2021 [73]	Refractory CRS post-FESS	128	90.0%	Azithromycin 250 mg 3 times per week	4 months
Lin Chien-Fu, 2020 [74]	CRSwNP post-FESS	126	100%	Clarithromycin 500 mg	3 months
de Oliveira, 2020 [75]	AERD	59	100%	Azithromycin 500 mg 3 times per week	3 months
Haxel, 2015 [37]	CRS post-FESS	67	55.2%	Erythromycin 250 mg	3 months
Amali, 2015 [76]	CRS post-FESS	66	42.4%	Azithromycin 250 mg	3 months
Perić, 2014 [77]	CRSwNP	80	100%	Clarithromycin 500 mg	2 months
Videler, 2011 [78]	CRS	60	52%	Azithromycin 500 mg per week	3 months
Wallwork, 2006 [41]	CRSsNP	64	0%	Roxithromycin 150 mg	3 months

CRS, chronic rhinosinusitis; CRSsNP, chronic rhinosinusitis without nasal polyps; CRSwNP, chronic rhinosinusitis with nasal polyps; FESS, functional endoscopic sinus surgery.

**Table 2 ijms-24-09489-t002:** Randomized controlled trials using low-dose long-term macrolides with comparison or add-on of intranasal corticosteroid sprays in chronic rhinosinusitis.

Reference	Patients	Number	Nasal Polyp	Macrolide	Duration	Comparison
Zeng, 2019 [79]	CRS post-FESS	205	70.0%	Clarithromycin 250 mg	3 months	Fluticasone propionate
Deng, 2018 [76]	CRS	74	56.8%	Clarithromycin 250 mg	3 months	Budesonide aqua nasal spray (add-on)
Varvyanskaya, 2014 [36]	CRSwNP post-FESS	66	100%	Clarithromycin 250 mg	3 months or 6 months	Mometasone furoate (add-on)
Korkmaz, 2014 [63]	CRSwNP	85	100%	Clarithromycin 250 mg	8 weeks	Mometasone furoate (add-on)
Zeng, 2011 [80]	CRSsNP	43	0%	Clarithromycin 250 mg	3 months	Mometasone furoate

CRS, chronic rhinosinusitis; CRSsNP, chronic rhinosinusitis without nasal polyps; CRSwNP, chronic rhinosinusitis with nasal polyps; FESS, functional endoscopic sinus surgery.

## Data Availability

Not applicable.

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
