# Peer review of "The Mechanism of Action and Clinical Efficacy of Low-Dose Long-Term Macrolide Therapy in Chronic Rhinosinusitis"

_ijms, 2023, doi:10.3390/ijms24119489_

Round 1
Reviewer 1 Report
Number of reference should be indicated in table 1 and 2 to facilitate reading.
Author Response
We appreciate the insightful and constructive comment from you. We have added the reference numbers in the tables.

Author Response
We thank the reviewer for their time and helpful comments to improve our manuscript. To follow your recommendation, a figure has added to summarize the contents of the text.
We have added a comment in the introduction as follows: Unless otherwise noted, all clinical studies were conducted on adult patients with CRS.
Based on your comments, we have added information to the text whether it is serum or local IgE.
3.3. Normal vs. high total IgE
Previous studies reported that only CRS patients with normal serum IgE levels (< 200 μg/L [41] or ≤ 250 U/ml [82]) benefited from LDLT macrolide treatment. However, the relationship between total serum IgE levels and LDLT macrolide treatment effects is still controversial. According to the studies that LDLT macrolide treatment was effective, the total serum IgE in the patient group was 188.63 ± 57.25 IU/mL [73] and 165.0 ± 195.2 μl/L [37]. Maniakas et al. reported that total serum IgE was higher in the azithromycin success group compared to the azithromycin failure group [91]. In addition, atopy status did not affect the clinical effect of clarithromycin in CRSsNP patients.[75]
Recently, of the 100 CRS patients who administered LDLT roxithromycin, 29 were determined to be responders [92]. Among clinical parameters, including nasal secretion and serum IgE, IL-5, blood eosinophil/neutrophil, allergy, asthma, nasal polyps, total IgE in nasal secretions was the only predictor of responder in multivariate models (odds ratio 4.76, 95% confidential interval 1.29-17.58). The authors suggest that local total IgE is a reliable biomarker instead of serum total IgE.
Reviewer 3 Report
nice review paper with time_consuming efforts
However, I could not find out neither conclusive author's opinion nor original future direction.
Thor
Through all manuscript, I feel very boring to read out previous papars result and discussion. I highly recommend authors to submit this paper to domestic English journals. It is because this review papar can be comprehensive for the strangers in this field.
Author Response
We appreciate your kind comment. There is a lot of previously published papers, so it seems that the content is not organized. To overcome this, a figure has added to summarize the contents of the text. Although focused on chronic rhinosinusitis, it can be applied to many airway diseases. We would be happy if the revised manuscript is more suitable for publication in this journal.

Reviewer 4 Report
Dear Editor,
thank you for the opportunity to read this summary of long term macrolide therapy. Overall, the current literature on this topic seems well summarized. To better illustrate the results, however, I would like a summary table or graph of the individual mechanisms.
Kind regards
Author Response
We thank the reviewer for the generous appraisal of our work. To follow your recommendation, a figure has added to summarize the contents of the text.

Reviewer 5 Report
This is a comprehensive review of the therapeutic effect of macrolide treatment of chronic rhinosinutis. The authors recognize the complexity of chronic rhinosinusitis. They present studies related to the interesting non-antibiotic effect of macroldes and critically summarize studies of clinical effect.
Author Response
We thank the reviewer for the generous appraisal of our work.
Round 2
Reviewer 3 Report
This review can be accepted for publication in MDPI journal, because it is conprehensive and better understandable for readers with good figures.
Reviewer 4 Report
Thank you for adding the figure to the manuscript.